# Assessment of Five-Foot Plantar Morphological Pressure Points of Children with Cerebral Palsy Using or Not Dynamic Ankle Foot Orthosis

**DOI:** 10.3390/children10040722

**Published:** 2023-04-13

**Authors:** Senem Guner, Serap Alsancak, Enver Güven, Ali Koray Özgün

**Affiliations:** Department of Prosthetics & Orthotics, Faculty Health of Science, Ankara University, 06290 Ankara, Turkey

**Keywords:** cerebral palsy (CP), DAFO (dynamic ankle-foot orthosis), plantar pressure distribution

## Abstract

People with spastic cerebral palsy (CP) often experience a decline in gait function and flexion. The children’s posture and hip strategy, which leads to knee flexion, predisposes these children to increased contact area in the medial foot region. This study investigated the use of DAFO (dynamic ankle-foot orthosis) prescribed to patients with cerebral palsy (CP) to determine the plantar pressure distribution with orthosis use. Eight children with spastic CP (age 4–12 years) were classified as Gross Motor Function Classification System (GMFCS) levels I-II with a maximum spasticity level of 3 in their ankle muscles according to the Modified Ashworth Scale. We assessed the plantar distribution by using eight WalkinSense sensors in each trial and exported data from the proprietary software (WalkinSense version 0.96, Tomorrow Options Microelectronics, S.A.). The plantar pressure distribution was conducted under two conditions: only shoes and DAFO with shoes. The activation percentages for sensor number 1 under the 1st metatarsal and sensor number 4 under the lateral edge of the heel were significantly different under the DAFO condition. The 1-point sensor activation percentage significantly decreased, while the 4-point sensor activation percentage increased during DAFO walking. According to our study findings, there was an increase in pressure distribution in the lateral part of the foot during the stance phase in DAFO. DAFO improved the gait cycle and influenced the plantar foot pressure in children with mild cerebral palsy.

## 1. Introduction 

Cerebral palsy (CP) is a complex pathology that affects cognition, language, sense, and gait patterns and severely impacts children [1]. Gait deviation is one of the most often problems in children with CP and causes motor disability. Muscle spasticity in the lower extremities may cause toe-in walking and crouch gait, resulting in a reduced ability to balance and falls during walking and ambulation training [2]. In patients with CP, spasticity, motor impairment, impaired balance, and muscle weakness can cause ankle and foot lever-arm dysfunction, which leads to foot deformity and gait disorders [3].

Diplegic gait is characterized as being a low-speed gait and is the main cause of muscle weakness and spasticity in the lower extremities [4]. Dynamic equinus seen due to spasticity in plantar flexor muscles is a very common deformity in both hemiplegic and diplegic patients with CP. A rigid ankle-foot orthosis (AFO), which provides passive control with the three-point pressure principle, is the most commonly prescribed and preferred for joint deformity control. The purpose of using AFO orthoses in children with CP is to reduce the pathological reflex pattern by positioning the peripheral joints and to create a more normal gait pattern by controlling the pathological movement of the joint [5]. In children with diplegic CP, the use of a rigid AFO can normalize the ankle function by limiting the ankle to a single neutral angle. The results showed that in the stance phase of the gait cycle, knee flexion improved and a decrease in using the required muscle strength [6].

A dynamic ankle-foot orthosis (DAFO) with its supra-malleolar type and underfoot support structure has been increasingly used as an alternative to AFO for patients with CP [7]. DAFO was developed based on the finding that the stimulation of the specific reflexogenic zone of the foot affects the more proximal muscles, and it has a special underfoot arch support structure [8]. Various AFOs can be used in rehabilitation processes. AFOs used in clinical practice vary widely and were characterized by their design, the materials used, and the stiffness of those materials. Any change in these three components will alter AFO’s control over the patient’s gait [9,10]. The factors affecting the rigidity of the AFO depend on several factors such as the mechanical properties of the material, trim lines, material thickness, and proximal structure shape [11,12]. AFOs control foot and ankle movements in different planes with their various designs. For example, solid AFO restricts ankle and foot movements in three planes, DAFO is more flexible and thinner and allows partial movement, PLS has a posterior malleolar trim line (posterior leaf spring), articulated HAFO allows dorsiflexion, and GRAFO is a single piece that supports the knee extension [13,14,15].

DAFO, supra-malleolar orthosis, is a very thin flexible orthosis designed to support and stabilize the forefoot, foot arches, and subtalar joint. The foot sole structure especially created for the patient is designed to reduce abnormal muscle activity and to affect biomechanical changes. The orthosis allows for very small movements in the midfoot and forefoot, supporting foot function and providing more proprioceptive feedback for posture and balance improvement. Improvement in motor skills increases with the presence of movement stability and postural security [7].

DAFO is designed as an orthosis that provides tone inhibition with a neutral alignment of the ankle and foot. A neutral forefoot in full contact with the orthosis and an outer structure that supports the subtalar joint is formed [16]. DAFO with free plantarflexion is a type of orthosis frequently used to facilitate the gross motor skills of children with cerebral palsy. The DAFO underfoot structure supports all the transverse, medial, and lateral longitudinal dynamic arches of the foot and reduces the pressure under the metatarsal heads and calcaneal fat pad [17]. It aims to effectively redistribute the pressure on the plantar surface of the foot to decrease the muscle tone of the foot [4,16,18] and control the equinus deformity. Bjornson et al. emphasized the early effect of the use of DAFO, especially on gross motor skills, in ambulatory children with cerebral palsy [19]. Condie and colleagues suggested that the goals of equinus management with the use of orthoses in children with CP can be divided into five distinct areas: preventing deformity, correcting deformity, promoting a base of support, facilitating skill training, and improving the efficiency of movement such as walking [20].

AFOs used in children with CP have a positive effect on gait kinematics and kinetics by supporting motor skills and reducing energy expenditure in walking [21,22,23,24]. Positive effects on walking include stride length, walking speed, active ankle range of motion, and maximum knee extension [25,26]. The use of AFO in hemiplegic CP decreased the mean plantarflexion in the stance phase and the maximum plantarflexion in the swing [27]. HAFO positively affected the passive ankle dorsiflexion angle in diplegic CP [28]. Lintanf et al.’s systematic review and meta-analysis reports indicated that in equinus gait who used posterior AFOs, these effects were increased in ankle dorsiflexion in the stance and swing phase of the gait cycle [29]. DAFO and a spring-like AFO provided near-normal ankle kinetic and kinematics motion in late stance and more normal push-off power at pre-swing.

With more dorsiflexion and knee flexion in the stance phase in the equinus pattern or dropped foot gait, DAFO is effective for children with a crouched gait pattern. Ankle dorsiflexion showed improvement in the swing and stance phase compared to barefoot, and it also did not differ in knee extension in stance, but hip extension was improved with DAFO [30,31].

Leunkeu et al. found that plantar pressure increases in the medial heel, decreases in the hallux and lateral heel during walking in hemiplegia, and increases the plantar loads on the first metatarsal, medial, and lateral midfoot in diplegia [32]. In children with diplegic and hemiplegic CP, the mean static plantar pressure was higher in the forefoot and midfoot regions and lower in the hindfoot region [33]. Aboutorabi et al. in their systematic review investigate the effect of AFO on gait pattern in children with CP; evidence shows an improvement in gross motor function with HAFO and SAFO in children with diplegic CP. They emphasized that it provides a more physiologically normal gait and restores ankle rockers. In the review study, it was determined that the studies generally used gait analysis for evaluation purposes and the pressure distribution under the foot was not included in the studies [30].

Gait analysis evaluation parameters have been proven to be a valuable method for assessing the effect of orthosis in CP. However, non-studies have used plantar pressure distribution in the assessment of gait when wearing DAFO. The aims of this study were the effects of DAFO orthotic treatments on the plantar pressure distribution during gait in spastic patients with spastic CP.

## 2. Methods

The evaluation was eight children with spastic diplegia CP (age 4–12 years); 2 were classified at Gross Motor Function Classification System (GMFCS) level I, and 6 were at GMFCS-level II; with a maximum spasticity level of 3 ankle muscles according to the Modified Ashworth Scale. The GMFCS objectively classifies a child’s current gross motor function in children with CP aged 1 to 12 years. The focus is on the child’s self-initiated movement with a particular emphasis on function in sitting and walking. The function is divided into five levels: children in Level I have the most independent motor function and children in Level V have the least [34]. The patients had had no surgical intervention in the previous twelve months and no botulism toxin injections in the last six months. Orthotic treatment was performed in the Ankara University Department of Prosthetics and Orthotics by the same experienced person.

Ethical approval was obtained for the study from the Medical Research Ethical Committee of the Ankara University Faculty of Medicine (approval no: 2022-370). All participants and their legal guardians were fully informed about the study and agreed to participate.

### 2.1. Orthoses Fabrication and Fitting

The manufacturers of the orthoses determined the specifications of the orthoses provided to each participant (including the type of DAFO) based on a physical examination performed before molding, specifically the range of motion, muscle tone, and walking barefoot (Table 1). As a result of the evaluation, inadequate ankle dorsiflexion and inadequate knee flexion or hyperextension was detected in participants, and all of them had equinus or flatfoot gait. Equinus was classified as excessive static or dynamic plantar flexion throughout the stance phase via visual observation of the child’s gait [35]. The same shoes designed for use with DAFO were worn with both orthoses (Figure 1). Of the children with CP, DAFO4 was applied to 2 of them, and DAFO3 was applied to 6 of them.

The dynamic ankle-foot orthosis has been produced from 3/32 in (0.24 cm) polypropylene with the use of a contoured footplate, which enclosed the dorsum of the forefoot, and a plantar flexion block, with dorsiflexion being allowed through the flexibility of the plastic and the absence of a calf strap. There was no hinge, ankle, and covered the posterior part of the leg to about 5–7.5 cm above the malleoli. The fact that DAFO is made of thin and flexible material prevents the orthosis from breaking. Additionally, DAFO has a tailor-made dynamic foot orthosis (DFO) section. This section contains medial and lateral longitudinal arch reinforcements in the foot and supporting reinforcement for the transverse arch and calcaneus. Gripping support is created in the medial and lateral longitudinal arches from the front of the calcaneus, and the inclination of the calcaneus is prevented. Thus, full contact with the orthosis with the foot is ensured, helping distribute and carry the body weight in a balanced way. Full contact support also positively affects tone inhibition. In addition, with the elevation of the toes in the orthosis, the toes flexors are stretched, while the 2nd, 3rd, and 4th toes are elevated in the orthosis. The elevation of the 1th toe was less than the others. We performed this elevation while the casting procedure plaster reduced and shaped the toes’ elevation; we did not need extra addition soft material (Figure 2). Thus, while the anterior transverse arch is supported, the metatarsal heads associated with them are placed on their beds (Figure 3). In our casting procedure, we created a natural footplate; we aligned the hindfoot, midfoot, and forefoot while allowing a natural arch to occur. The calcaneus is brought forward to achieve dorsiflexion at the talocrural joint. We checked the sinus tarsi pressure on the plaster. Correction of forefoot dorsiflexion proximal to the base of the 5th was used to protect the integrity of the midfoot. 

With the velcro band surrounding the ankle at an angle of 45°, displacement of the calcaneus and the dorsal band of the foot within the beds of the metatarsals are prevented. With all these features, the flexible supra malleolar structure allows controlled movement of the subtalar joint and allows the foot to perform adduction, inversion, and abduction eversion movements within the orthosis, but this does not cause any adverse effects on mediolateral stabilization (Figure 4).

### 2.2. Plantar Pressure Distribution 

The plantar pressure distribution under foot measurements was performed using 8 WalkinSense sensors in each trial and exported from the proprietary software (WalkinSense version 0.96, Tomorrow Options Microelectronics, S.A.) [36]. The system consisted of a data acquisition and processing unit and individual sensors attached to the participants’ socks to measure the plantar pressure and activation percentage. The participants wore standard socks, which were fitted with the WalkinSense sensors and provided to the participants. Sensor number 1 was placed under the 1st metatarsal, sensor number 2 was placed under the 5th metatarsal, sensor number 3 was placed on the lateral of the midfoot, sensor number 4 was placed under the lateral of the heel, and sensor number 5 was placed under the medial heel (Figure 5). The order of the trials without a DAFO and with DAFO was counterbalanced in this study. Foot plantar pressures were recorded during dynamic walking in patients with CP. In this study, a statistical evaluation was made by taking the pressure values in the mid-stance phase. 

Descriptive statistics were calculated for all the data. Statistical analyses were performed using the SPSS version 23 software. The nonparametric Wilcoxon test was used to compare only shoe measurements and DAFO with shoes, respectively, within each subject. Homogeneity could not be achieved within the group because it was studied with a small group. The statistical significance level was set at *p* < 0.05.

## 3. Results

Eight children with mild cerebral palsy (age = 9 ± 3.3 years; weight = 31 ± 10.2 kg; height = 126.5 ± 19.2 cm), who all had a dominant right leg, were recruited and performed the short distance stride repetitions while wearing and removing DAFO. All of them had received physical therapy at the rehabilitation center. The activation percentages for sensors 1 and 4 were significantly different under the DAFO conditions (Table 2). The 1-point sensor activation percentage significantly decreased, while the 4-point sensor activation percentage increased during DAFO walking. In the plantar pressure distribution that we obtained without DAFO walking in children with CP, the activation percentage is higher at pressure points 1, 2, and 3. This shows us that the contact of the 1st and 5th metatarsal phalanx points during the stance phase is more than the other points; we see that the medial–lateral longitudinal part of the foot is used more actively. The increase in the percentage of activation point 4 in the stance phase of gait with the use of DAFO shows that the activation of the lateral part of the foot increases. With the use of DAFO, we see that the weight transfer in the lateral part of the foot becomes more active with the shift of the maximum pressure point from two to three times.

## 4. Discussion 

This study aimed to evaluate the effect of DAFOs on foot plantar pressure distribution in patients with CP. DAFOs’ limited plantar flexion at push-off affects plantar pressure distribution underfoot. Our results showed an increase in the lateral foot pressure distribution at the stance phase in the DAFO. Orthotic devices are invented to support the ankle, correct deformities, and prevent further occurrences. With the DAFO arch support, the weight is evenly distributed throughout the foot, and the structure that stimulates the foot reflexes creates an effect closer to the normal function. Mediolateral stabilization provides grade dorsiflexion better and has been proven to reduce abnormal plantarflexion [37]. Many studies have indicated that the increased plantar flexor moment in the stance phase provides some biomechanical benefits during walking to a child with spastic diplegia. They concluded that AFO improves gait function in diplegic CP compared with barefoot walking [38,39]. Buckon et al. reported HAFO, PLS, and SAFO to decrease knee hyperextension by preventing ankle plantarflexion [40]. On the other hand, researchers found gait conditions with different orthoses (barefoot, HAFO, and DAFO) in patients with spastic hemiplegic CP. They emphasized that the use of orthosis especially affects the spatiotemporal parameters, and there was a significant increase in both stride and stride length, but there was no difference between the designs [41]. Schwarze et al. showed patients with CP DAFOs and modular shank supply positively affected gait indices and increased step length, velocity, and decreased cadence. They stated that the kinematic effects were mostly in the ankle and knee joints [42]. Lam et al. reported that orthoses provide control over initial contact positioning and plantarflexion in the swing phase as a result of gait analysis and electromyographic examination in patients with CP dynamic equinus using AFOs and DAFOs. The DAFOs allowed a significantly larger total ankle range of motion than the AFOs [17]. For children with apparent equinus, a decrease in ankle power generation during push-off and a better ankle position during the swing, as well as a decrease in knee flexion during terminal stance, were observed. The use of AFO can prevent the midfoot brake, which leads to a correct roll-off that increases the lever arm, resulting in a correspondingly high plantar flexion moment [43]. There is strong evidence for changes in biomechanical parameters during gait, particularly an increase in ankle dorsiflexion value at initial contact and during a swing, using posterior AFOs in children with equinus gait, and if the goal is to improve ankle kinematics in a child with unilateral or bilateral cerebral palsy, dynamic AFO is indicated to be appropriate [29].

Many factors that play a role in the development of equinus deformity in patients with CP, including muscle imbalance, abnormal loading patterns, rotational gait deviations, and intrinsic anatomical features probably all play a role in the development of these deformities, and the plantar flexor lever arm is shortened in this foot deformity, which is often compensated by hip and knee flexion, and joint position affects gait control [44]. In the pes planovalgus, the foot is not in the same plane with the progression line, and this leads to insufficient movements. In addition, this deformity shows the hindfoot valgus, forefoot abduction, forefoot supination, and medial longitudinal arch collapse [3]. Adequate correction of foot deformity and/or the use of orthoses can improve the foot function as adequate loading patterns. Since the special underfoot structure of DAFO affects the foot kinematics, we think that it will affect the pressure distribution on the plantar surface of the foot during walking. An increase in foot kinematics, especially in foot dorsiflexion, during walking can increase plantar pressure in the hindfoot. This situation may provide positive effects in gait in children with CP who have equinus gait.

Plantar pressure measurements clearly reflect changes in foot postural alignment and can be used to detect toe walking in children with CP [45]. Femery et al. pointed out that children with hemiplegia show marked differences in the plantar load distribution between the affected and unaffected extremities, particularly for the first metatarsal head and midfoot [46]. Some studies have found that static plantar pressure is higher in the forefoot and midfoot regions and lower in the hindfoot region in children with diplegic and hemiplegic CP [32,33]. Our study findings showed that in the evaluation of plantar pressure distribution under the foot without DAFO in spastic diplegic children, more weight was given especially to the forefoot and midfoot parts of the foot. We evaluated only the effect of DAFO on the pressure distribution in the foot, and we found that it affects the pressure in the foot in diplegic children with CP, distributes the pressure evenly under the foot, and increases it in the lateral heel.

The effect of DAFO on gait patterns has been shown in some studies, but we have not found any study on how it affects the pressure distribution under the foot. The use and application of gait analysis are extensive and can take time, and measurements taken with plantar pressure can be applied more practically in clinical applications and can be adapted to various areas. With the plantar pressure evaluation, it is possible to obtain information about abnormal gait, the severity of foot diseases and deformities, and rehabilitation status [47]. Jolanta et al. investigated plantar pressure and spatiotemporal parameters in spastic diplegia children with and without AFO, and they emphasized that a significant difference was found in the toes, the metatarsal heads, the medial arch, and the heel compared with the control group. They stated that the pressure decreased particularly in the heel in diplegic children with CP, and they did not detect any difference in the distribution of plantar pressure with and without AFO [48]. However, the results of our study show that plantar pressure changes in children using DAFO. In our study, it was found that when DAFO was used in children with CP, the pressure distribution in the foot changed and spread more evenly under the feet. It has been observed that the use of DAFO reduces the pressure under the foot in the forefoot and increases the activation of the hindfoot. Balanced pressing and distribution under the feet will positively affect the walking pattern of children with CP. In our study, DAFO involves arch reinforcement under the foot, under-toe elevation reinforcements in the positive cast, and calcaneus bedding provide tone inhibition. It has been observed that the pressure distribution under the foot is transferred from the anterior part of the foot to the hind and midfoot. In children with pes planovalgus and equinovarus, this effect supports gait stability in the stance phase and contributes to controlling excessive flexion or extension in the knee and hip joint by controlling plantar flexion of the ankle.

This study has several limitations. First, due to the small number of individuals with CP and the heterogeneity of the participants, the power of the statistical tests was generally low. Second, the long-term effects of an orthosis on functional activity also need to be investigated. Third, its effect on underfoot pressure distribution is comparable to other orthotic designs.

## 5. Conclusions

In the current study, the findings of plantar pressure distribution indicated the potential use of DAFO orthosis to improve the gait cycle and influence foot motion in children with mild cerebral palsy. 

## Figures and Tables

**Figure 1 children-10-00722-f001:**
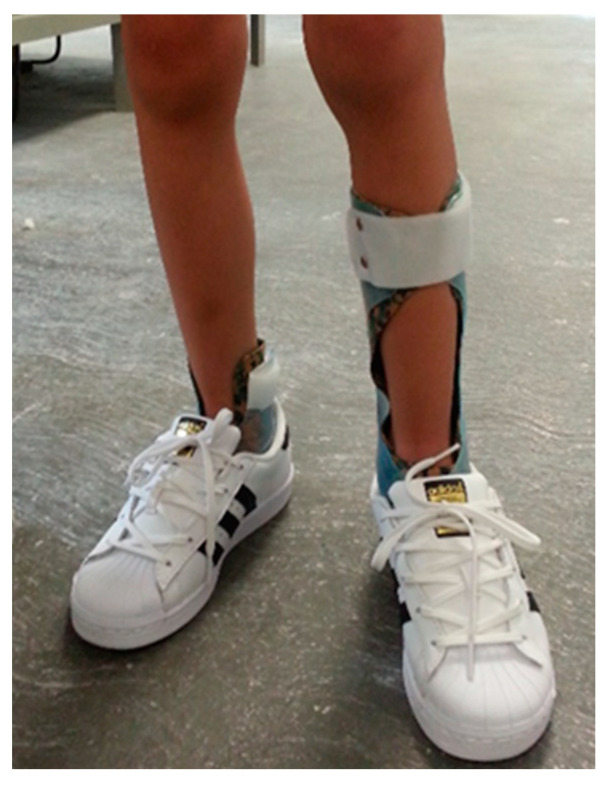
Application of DAFO4 and DAFO Turbo (provides strength and stability, blocking plantarflexion and dorsiflexion) on the patient.

**Figure 2 children-10-00722-f002:**
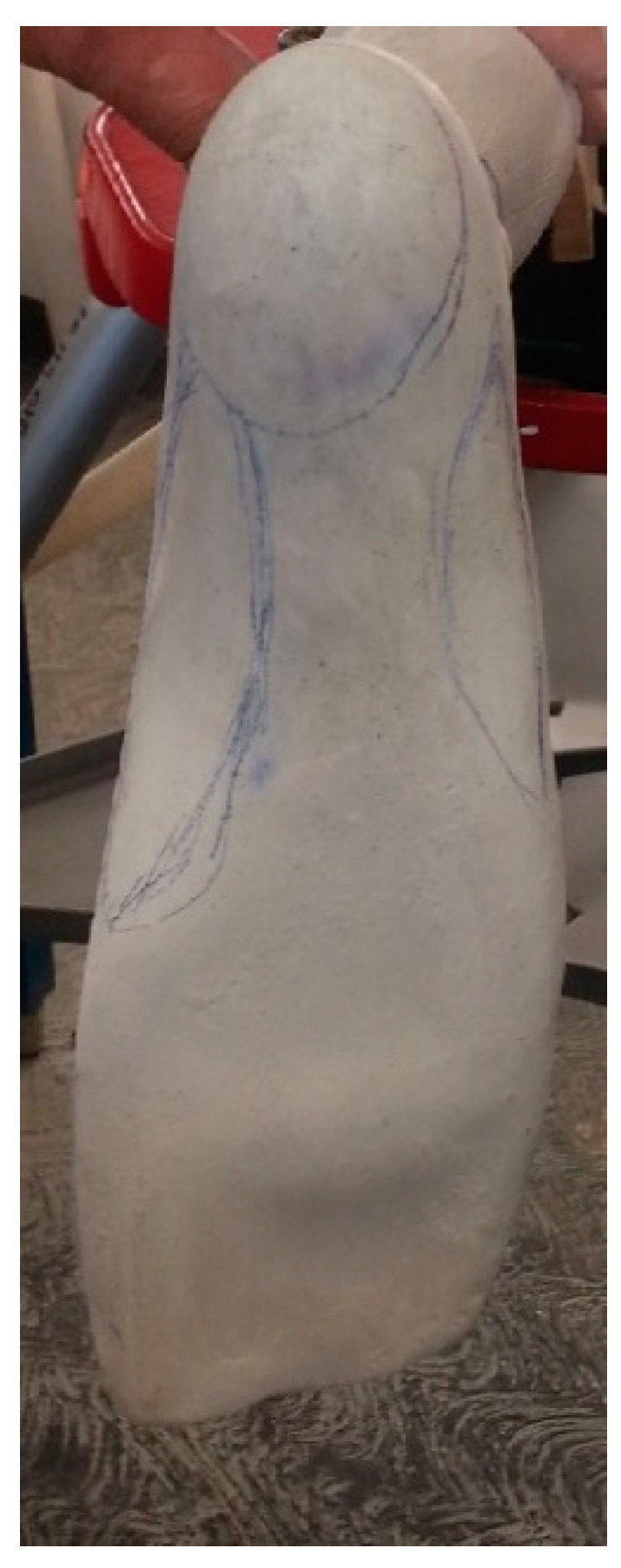
Demonstration of the plantar surface on the plaster model.

**Figure 3 children-10-00722-f003:**
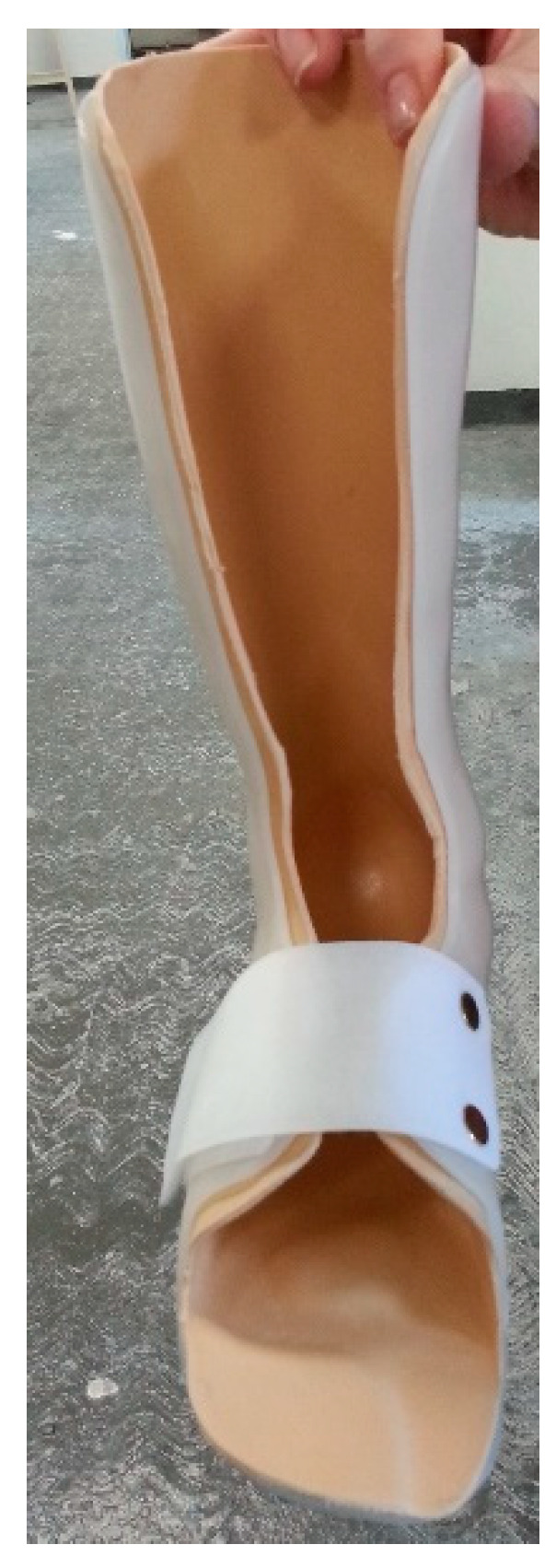
The design of DAFO3 (blocks plantarflexion, allows free dorsiflexion, and provides additional medial/lateral stability).

**Figure 4 children-10-00722-f004:**
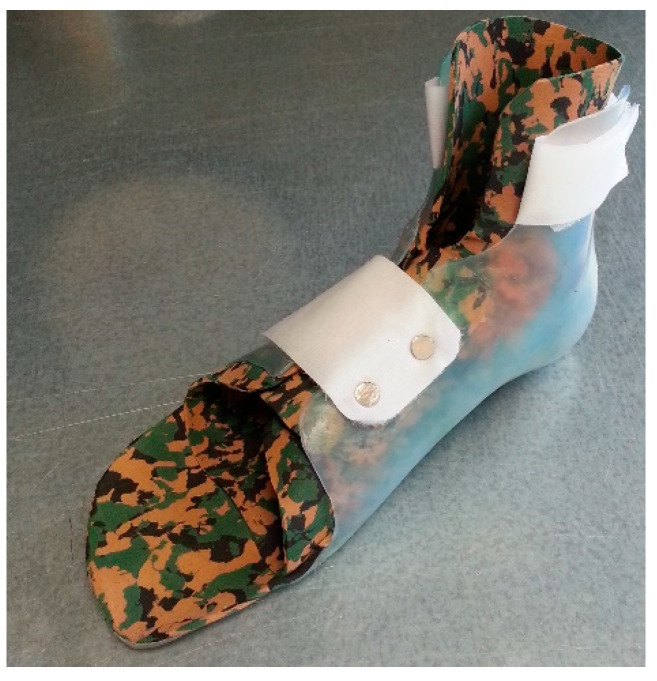
The design of DAFO 4 with a posterior strap (for flexible control of foot position and alignment, allows full plantar flexion and dorsiflexion).

**Figure 5 children-10-00722-f005:**
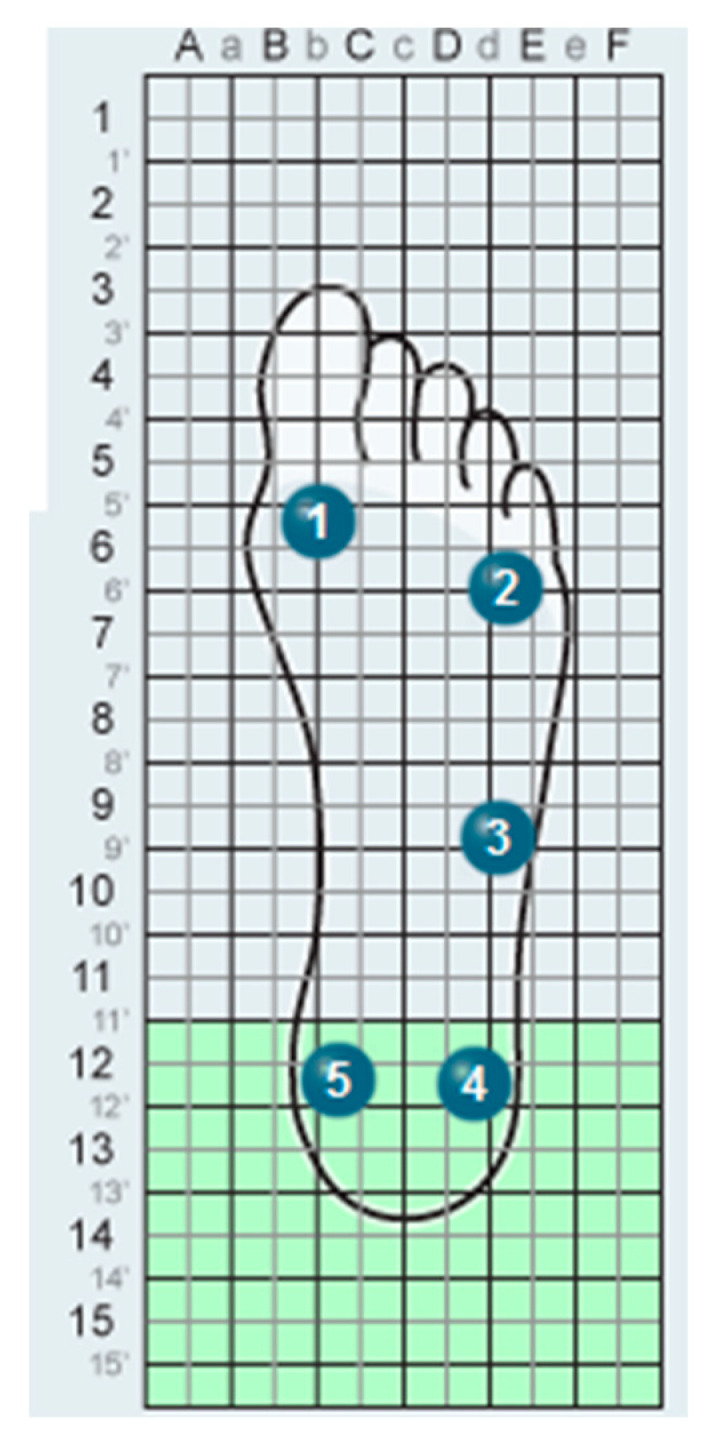
Underfoot WalkinSense sensors placed a screenshot.

**Table 1 children-10-00722-t001:** Patient information. M, male; F, female; ROM, range of motion; (°) degree.

Case	Age	Sex	Diagnosis	Type of Orthosis	Total Ankle ROM (Extended Knee)	Total Ankle ROM (Flexed Knee)	İnitial Contact (without Orthosis)	İnitial Contact (with the Orthosis)	Max Knee Extension in the Stance Phase
1	12	M	Spastic Diplegia	DAFO3	30°	38°	İnitial ground contact made with forefoot	İnitial ground contact made with both the forefoot and hindfoot	Hyperextension
2	5	F	Spastic Diplegia	DAFO4	48°	55°	İnitial ground contact made with both the forefoot and hindfoot	İnitial ground contact made with both the forefoot and hindfoot	Not Hyperextension
3	12	M	Spastic Diplegia	DAFO3	35°	42°	İnitial ground contact made with forefoot	İnitial ground contact made with both the forefoot and hindfoot	Hyperextension
4	4	M	Spastic Diplegia	DAFO3	40°	47°	İnitial ground contact made with both the forefoot and hindfoot	İnitial ground contact made with both the forefoot and hindfoot	Hyperextension
5	10	F	Spastic Diplegia	DAFO3	34°	40°	İnitial ground contact made with forefoot	İnitial ground contact made with both the forefoot and hindfoot	Hyperextension
6	9	M	Spastic Diplegia	DAFO4	47°	55°	İnitial ground contact made with both the forefoot and hindfoot	İnitial ground contact made with both the forefoot and hindfoot	Not Hyperextension
7	9	M	Spastic Diplegia	DAFO3	37°	45°	İnitial ground contact made with forefoot	İnitial ground contact made with both the forefoot and hindfoot	Hyperextension
8	11	F	Spastic Diplegia	DAFO3	32°	38°	İnitial ground contact made with forefoot	İnitial ground contact made with both the forefoot and hindfoot	Hyperextension

**Table 2 children-10-00722-t002:** Demonstration plantar pressure distributions on 1, 2, 3, 4 and 5 sensors maximum pressure and sensor activation % with wear-only shoes and DAFO with shoes in the stance phase (Wilcoxon signed ranks test, median values).

Stance Phase Plantar Pressure Distributions	Only Shoes	DAFO with Shoes	*p*
Maximum pressure kg/cm^2^	1.99	1.71	0.123
Maximum pressure sensor	2	3	0.037 *
1 sensor activation %	92	24.5	0.028 *
2 sensor activation %	89.2	56	0.161
3 sensor activation %	81	88	0.161
4 sensor activation %	58	90	0.036 *
5 sensor activation %	60	80	0.161

**p* < 0.05, statistically significant result.

## Data Availability

The datasets generated or analyzed during the current study are available from the corresponding author upon reasonable request.

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
