# Peer review of "Assessment of Five-Foot Plantar Morphological Pressure Points of Children with Cerebral Palsy Using or Not Dynamic Ankle Foot Orthosis"

_children, 2023, doi:10.3390/children10040722_

Round 1

Reviewer 1 Report

This study compares plantar pressure during walking in 8 children with cerebral palsy under two conditions: wearing dynamic ankle foot orthoses and barefoot.

The introduction gives a thorough description of AFOs and what they are capable of, citing evidence for their effectiveness. It demonstrates that this study fills a gap in the literature. However, it would also be useful to review the literature on analyses of gait patterns in CP. This would help the authors to interpret their plantar pressure findings (e.g., if the gait analysis literature shows that AFOs improve heel strike, then you would expect to see high pressure readings on the sensors positioned on the heel at that phase of the gait cycle).

Ethics approval: “Ethical approval was given by the ethics committee of the 92 university's faculty of medicine.” Please provide the name of the Ethics committee and the project approval number.

Regarding the designs of the DAFOs used in this study: Two photos are provided of DAFO 3 and DAFO 4. What did DAFO 1 and DAFO 2 look like? How many children in the study used each type? And did you examine any differences between the types? (That may not be appropriate with the small sample size.)

Please clarify the footwear in the two conditions. In some places, it says that the no-DAFO condition was “barefoot” but elsewhere it says socks were worn. In the DAFO condition, did they wear shoes as well as DAFOs (as in Figure 1)? If so, then differences might be partly attributable to the shoes, mightn’t they?

Did all children wear 2 DAFOs? I somehow had the impression that they had diplegia, but now I look back, it doesn’t seem to specify whether they had diplegia or hemiplegia.

Table 1 is missing, so I cannot assess the results of the study. If this table were shown, I would be looking for the mean, SD, and range at each sensor point at each phase of the gait cycle, so that I can judge whether there are trends for any of the sensors other than 1 and 4, and in what direction. With the very small n, there may be important differences that don’t reach significance. The authors interpret their results as indicating more pressure on the lateral part of the foot, but the results from sensors 1 and 4 might indicate more pressure on the hindfoot and less on the forefoot.  

The Discussion compares the results with those of previous studies. It is hard to assess, without seeing the results of the present study.

There are significant problems with English expression throughout this paper. The following are corrections only to the Abstract. I would recommend that somebody with good written English skills should read through the entire paper and make corrections as needed.

·         Change: “People with Spastic cerebral palsy (CP) often experiences” to “People with spastic cerebral palsy (CP) often experience”.

·         Please describe what “antepulsion posture” is when walking or use a different term. When I googled this term I found it used in other European languages, but very little in English. I’m not sure exactly which phase(s) of the gait cycle it encompasses.

·         Change “flexion, these children” to “flexion. These children”.

·         Please change “Eight spastic CP (age 4-12 years) patients” to “Eight children with spastic CP (age 4-12 years)”.

·         Please change “of level I-II according to at Gross Motor Function Classification System (GMFCS)” to “classified as Gross Motor Function Classification System (GMFCS) levels I-II”.

·         Please change “level of 3 ankle muscles” to “level of 3 in their ankle muscles”.

·         Please change “lateral of the heel” to “lateral edge of the heel”.

·         Please change “an increase in lateral part of foot pressure distribution at the stance phase” to “an increase in pressure distribution in the lateral part of the foot during the stance phase”.

Author Response

Thank you for your comments.

This study compares plantar pressure during walking in 8 children with cerebral palsy under two conditions: wearing dynamic ankle foot orthoses and barefoot.

The introduction gives a thorough description of AFOs and what they are capable of, citing evidence for their effectiveness. It demonstrates that this study fills a gap in the literature. However, it would also be useful to review the literature on analyses of gait patterns in CP. This would help the authors to interpret their plantar pressure findings (e.g., if the gait analysis literature shows that AFOs improve heel strike, then you would expect to see high pressure readings on the sensors positioned on the heel at that phase of the gait cycle).

Necessary information and additions have been made.

Ethics approval: “Ethical approval was given by the ethics committee of the 92 university's faculty of medicine.” Please provide the name of the Ethics committee and the project approval number.

Necessary information and additions have been made.

Regarding the designs of the DAFOs used in this study: Two photos are provided of DAFO 3 and DAFO 4. What did DAFO 1 and DAFO 2 look like? How many children in the study used each type? And did you examine any differences between the types? (That may not be appropriate with the small sample size.)

There is no DAFO 1 model and the model using the DAFO 2 ankle joint structure was not used in the study. This is a classification in itself, basically, the underfoot structure of the orthosis does not change.

Please clarify the footwear in the two conditions. In some places, it says that the no-DAFO condition was “barefoot” but elsewhere it says socks were worn. In the DAFO condition, did they wear shoes as well as DAFOs (as in Figure 1)? If so, then differences might be partly attributable to the shoes, mightn’t they?

Correction added.

Did all children wear 2 DAFOs? I somehow had the impression that they had diplegia, but now I look back, it doesn’t seem to specify whether they had diplegia or hemiplegia.

Necessary information and additions have been made.

Table 1 is missing, so I cannot assess the results of the study. If this table were shown, I would be looking for the mean, SD, and range at each sensor point at each phase of the gait cycle, so that I can judge whether there are trends for any of the sensors other than 1 and 4, and in what direction. With the very small n, there may be important differences that don’t reach significance. The authors interpret their results as indicating more pressure on the lateral part of the foot, but the results from sensors 1 and 4 might indicate more pressure on the hindfoot and less on the forefoot. 

Table 1 has been added, results and percentages are in the table.

The Discussion compares the results with those of previous studies. It is hard to assess, without seeing the results of the present study.

There are significant problems with English expression throughout this paper. The following are corrections only to the Abstract. I would recommend that somebody with good written English skills should read through the entire paper and make corrections as needed.

Necessary corrections have been made.

  • Change: “People with Spastic cerebral palsy (CP) often experiences” to “People with spastic cerebral palsy (CP) often experience”.

Necessary corrections have been made.

  • Please describe what “antepulsion posture” is when walking or use a different term. When I googled this term I found it used in other European languages, but very little in English. I’m not sure exactly which phase(s) of the gait cycle it encompasses.

Necessary corrections have been made.

  • Change “flexion, these children” to “flexion. These children”.

Necessary corrections have been made.

  • Please change “Eight spastic CP (age 4-12 years) patients” to “Eight children with spastic CP (age 4-12 years)”.

Necessary corrections have been made.

  • Please change “of level I-II according to at Gross Motor Function Classification System (GMFCS)” to “classified as Gross Motor Function Classification System (GMFCS) levels I-II”.

Necessary corrections have been made.

  • Please change “level of 3 ankle muscles” to “level of 3 in their ankle muscles”.

Necessary corrections have been made.

  • Please change “lateral of the heel” to “lateral edge of the heel”.

Necessary corrections have been made.

  • Please change “an increase in lateral part of foot pressure distribution at the stance phase” to “an increase in pressure distribution in the lateral part of the foot during the stance phase”.

Necessary corrections have been made.

Reviewer 2 Report

The paper provides an evaluation of the qualities of an orthosis, the DAFO, in eight children aged 4 to 12 years with cerebral palsy. To introduce anything that can improve the quality of life of these children is important.

However, the limitations of the paper as an "article" according to the definitions of "Children" itself (https://www.mdpi.com/journal/children/instructions#submission ) are very evident and arise from several points, among others, the limitations of the cerebral palsy characteristics of the eight children presented as the "sample" - which is not at all homogeneous - and the characteristics of the equipment.

As pointed out in the limitations of the work (lines 248 …) “This study has several limitations. First, only 8 children with CP with a mix of crouch and/or equinus gait participated. Because of the small sample size and heterogeneity of the participants, the power of the statistical tests was generally low.”

The problem of the authors is not only that the sample is small, it is that the sample is heterogeneous. Heterogeneity not described by the authors. The authors only mention "a mix of crouch and/or equinus gait". Therefore, there is no statistic that can describe the value of the data founded.

Parallel, there is a limitation arising from the equipment. The experiment just records 5 points of contact during the stance phase and not a complete plantar pressure. The knowledge of the force in the complete surface of contact (complete pressure) is important, in special, if we have “a mix of crouch and/or equinus gait”.

In general conclusion: As stated at the beginning, “To introduce anything that can improve the quality of life of these children is important“ but the study has to be completely redesigned,. for example, as a pilot study. Analysing the treatment of each child's data. Second, specify that the equipment (very practical to use and useful according to the purposes for which it was designed) is not for a complete plantar pressure, but just for the study of "5 foot supports” during the stance phase".

Those limitations greatly limit the admission of the work as an “article”

The quality and impact of the data study is not in accordance with https://www.mdpi.com/journal/children/instructions#submission

However if the study will be redesigned and in order to facilitate understanding for those who are not specialised in the assessment of cerebral palsy using the GMFCS scale: Important a brief note that the GMFCS scale has five levels, I and II being the lowest. Quote that is for this reason the text designated as "mild" the characteristic of the eight participants in the sampleAlso indicate what is the qualifications of the person that attributed this classification of “mild” or, I and II, based on the GMFCS five levels

Line 235: What the meaning of "mild centres of gravity

Line 186: Precise: “Many studies stated that the increased plantar flexor moment in the  stance phase was a biomechanical utility of orthoses in spastic diplegia”.

Author Response

Thank you for your comments.

The paper provides an evaluation of the qualities of an orthosis, the DAFO, in eight children aged 4 to 12 years with cerebral palsy. To introduce anything that can improve the quality of life of these children is important.

However, the limitations of the paper as an "article" according to the definitions of "Children" itself (https://www.mdpi.com/journal/children/instructions#submission ) are very evident and arise from several points, among others, the limitations of the cerebral palsy characteristics of the eight children presented as the "sample" - which is not at all homogeneous - and the characteristics of the equipment.

Added information about the homogeneity of the study.

As pointed out in the limitations of the work (lines 248 …) “This study has several limitations. First, only 8 children with CP with a mix of crouch and/or equinus gait participated. Because of the small sample size and heterogeneity of the participants, the power of the statistical tests was generally low.”

The problem of the authors is not only that the sample is small, it is that the sample is heterogeneous. Heterogeneity not described by the authors. The authors only mention "a mix of crouch and/or equinus gait". Therefore, there is no statistic that can describe the value of the data founded.

Parallel, there is a limitation arising from the equipment. The experiment just records 5 points of contact during the stance phase and not a complete plantar pressure. The knowledge of the force in the complete surface of contact (complete pressure) is important, in special, if we have “a mix of crouch and/or equinus gait”.

Measurements are made by placing sensors at certain points on the underfoot pressure measuring device used. Unfortunately, we cannot evaluate the entire bottom of the foot with this device.

In general conclusion: As stated at the beginning, “To introduce anything that can improve the quality of life of these children is important“ but the study has to be completely redesigned,. for example, as a pilot study. Analysing the treatment of each child's data. Second, specify that the equipment (very practical to use and useful according to the purposes for which it was designed) is not for a complete plantar pressure, but just for the study of "5 foot supports” during the stance phase".

The title of the work has been changed. It was stated that it was a pilot study.

Those limitations greatly limit the admission of the work as an “article”

The quality and impact of the data study is not in accordance with https://www.mdpi.com/journal/children/instructions#submission

However if the study will be redesigned and in order to facilitate understanding for those who are not specialised in the assessment of cerebral palsy using the GMFCS scale: Important a brief note that the GMFCS scale has five levels, I and II being the lowest. Quote that is for this reason the text designated as "mild" the characteristic of the eight participants in the samples indicate what is the qualifications of the person that attributed this classification of “mild” or, I and II, based on the GMFCS five levels

Necessary information and additions have been made.

Line 235: What the meaning of "mild centers of gravity

Chow Tong-Hsien. Traceable Features of Static Plantar Pressure Characteristics and Foot Postures in College Students with Hemiplegic Cerebral Palsy. J. Pers. Med. 2022;12: 394.

Chow et al. defined in this way.

Line 186: Precise: “Many studies stated that the increased plantar flexor moment in the  stance phase was a biomechanical utility of orthoses in spastic diplegia”.

Reviewer 3 Report

The manuscript shows the DAFO utilizations in children with cerebral palsy. I have some considerations for you.

Add authors affiliations like the template rules.

Abstract: there is no need to write the words “purpose, methods, results”.

Introduction

Line 32: please add a full stop after [3].

Line 39: although this time is plural “AFO” abbreviation has already been explained in line 34.

Same comment for DAFO in lines 37, 45, 58 and 61.

Line 50: (1989) there is no need to write the year because there are not the mdpi citation rules.

Line 65: reference 14 must be corrected (square brackets).

Line 67 and 73: a full stop is always after “al”.

Methods

There is a lack of information about how did you collect data? When? Where? How many people are involved on this procedure?

Ethical committee and an informed consent must be given to participants or legal tutors in this case.

Results

Numeral data, percentages must be added in this section. This section must be rewritten.

I cannot see table 1.

Discussion

Line 186: correct reference 20 (square brackets).

Line 192, 215, 220, 222 and 231: full stop after “al” should be added.

Line 236: a square bracket should be added.

Discussion has a lot of comparisons between studies to support the results.

Author Response

The manuscript shows the DAFO utilizations in children with cerebral palsy. I have some considerations for you.

Thank you for your comments.

Add authors affiliations like the template rules.

Abstract: there is no need to write the words “purpose, methods, results”.

Necessary corrections have been made.

Introduction

Line 32: please add a full stop after [3].

Necessary corrections have been made.

Line 39: although this time is plural “AFO” abbreviation has already been explained in line 34.

Necessary corrections have been made.

Same comment for DAFO in lines 37, 45, 58 and 61.

Necessary corrections have been made.

Line 50: (1989) there is no need to write the year because there are not the mdpi citation rules.

Necessary corrections have been made.

Line 65: reference 14 must be corrected (square brackets).

Necessary corrections have been made.

Line 67 and 73: a full stop is always after “al”.

Necessary corrections have been made.

Methods

There is a lack of information about how did you collect data? When? Where? How many people are involved on this procedure?

Necessary information and additions have been made.

Ethical committee and an informed consent must be given to participants or legal tutors in this case.

Necessary information and additions have been made.

Results

Numeral data, and percentages must be added in this section. This section must be rewritten.

Table 1 has been added, results and percentages are in the table.

I cannot see table 1.

Discussion

Line 186: correct reference 20 (square brackets).

Necessary corrections have been made.

Line 192, 215, 220, 222 and 231: full stop after “al” should be added.

Necessary corrections have been made.

Line 236: a square bracket should be added.

Necessary corrections have been made.

Discussion has a lot of comparisons between studies to support the results.

Round 2

Reviewer 1 Report

This paper reports a study of 8 children with spastic diplegia whose plantar pressure was compared while walking in a dynamic AFO versus no AFO.

This is the second round of review. Some of the changes I requested last time have been made. In this review, I will only focus on the ones that are still outstanding.

Regarding the designs of the DAFOs used in this study: Could you please specify how many children wore DAFO 3 and how many wore DAFO 4?

In the previous version, the results table was is missing. It has been added in this version, but the table does not include any indication of spread (SD and range). The results appear to show that the pressure is higher in the hindfoot (sensors 4 and 5) and lower in the forefoot (sensors 1 and 2) at midstance. However, the authors interpret their results as indicating more pressure on the lateral part of the foot. It is difficult to find support for this interpretation in Table 1. (However, I note that in the Discussion, in relation to the results of Leunkeu et al. and Galli et al., the authors do note that they got more weight on the forefoot and midfoot areas without DAFOs in the present study.)

Do the authors consider it relevant to look at the plantar pressure at points in the gait cycle other than midstance? The Discussion begins by discussing the effects of DAFOs on push-off.

The following corrections are still needed in the Abstract.

- Last time I asked for a change from: “People with Spastic cerebral palsy (CP) often experiences” to “People with spastic cerebral palsy (CP) often experience”. – This has not been done. It still says “experiences” not “experience”.

 - Last time I asked for a change from: “flexion, these children” to “flexion. These children”. – This punctuation change has not been made.

 - Please change “years) patients of classified” to “years) classified”. In other words, omit “patients of”.

 In my previous review, I noted that there are significant problems with English expression throughout this paper and I recommended that somebody with good written English skills should read through the entire paper and make corrections as needed. This has not been done.

Author Response

Thank you for your comments.

This paper reports a study of 8 children with spastic diplegia whose plantar pressure was compared while walking in a dynamic AFO versus no AFO.

This is the second round of review. Some of the changes I requested last time have been made. In this review, I will only focus on the ones that are still outstanding.

Regarding the designs of the DAFOs used in this study: Could you please specify how many children wore DAFO 3 and how many wore DAFO 4?

-Of the children with CP, DAFO 4 was applied to 2 of them and DAFO 3 was performed to 6 of them

In the previous version, the results table was is missing. It has been added in this version, but the table does not include any indication of spread (SD and range). The results appear to show that the pressure is higher in the hindfoot (sensors 4 and 5) and lower in the forefoot (sensors 1 and 2) at midstance. However, the authors interpret their results as indicating more pressure on the lateral part of the foot. It is difficult to find support for this interpretation in Table 1. (However, I note that in the Discussion, in relation to the results of Leunkeu et al. and Galli et al., the authors do note that they got more weight on the forefoot and midfoot areas without DAFOs in the present study.)

-We used the nonparametric Wilcoxon Signed Ranks Test and gave Median values.

-According to our results, the 4th and 1st number of sensor activation % are statistically significant, the 3rd and 5th activation percentages increased with the use of DAFO, although it was not statistically significant. In addition, the sensor pressure was found to be the highest in the 3rd sensor in the use of DAFO. We took out the places where we wrote the middle foot. We change it.

Do the authors consider it relevant to look at the plantar pressure at points in the gait cycle other than midstance? The Discussion begins by discussing the effects of DAFOs on push-off.

-It can be looked at, but we thought that the mid-stance phase would give clearer results. The most effective limitation of DAFO in general is to restrict plantarflexion, which affects the push of phase. The effect in push-off is very obvious, we wanted to focus on pressure and activation in the mid-stance phase, where full weight is placed on the foot.

The following corrections are still needed in the Abstract.

- Last time I asked for a change from: “People with Spastic cerebral palsy (CP) often experiences” to “People with spastic cerebral palsy (CP) often experience”. – This has not been done. It still says “experiences” not “experience”.

Necessary corrections have been made.

 - Last time I asked for a change from: “flexion, these children” to “flexion. These children”. – This punctuation change has not been made.

-Necessary corrections have been made.

 - Please change “years) patients of classified” to “years) classified”. In other words, omit “patients of”.

-Necessary corrections have been made.

 In my previous review, I noted that there are significant problems with English expression throughout this paper and I recommended that somebody with good written English skills should read through the entire paper and make corrections as needed. This has not been done.

-We edited in English in the Pooltext program.

Reviewer 2 Report

The study under review deserves my appreciation. The authors' replies show an effort to respond to the warnings, but the work still needs some fine-tuning, whether in the presentation of the results (a table with all the results is suggested below) or in some conceptual (?) or terminological (?) adjustments that are always difficult for non-native speakers of English to express.

1. About the new title

The authors accepted the suggestion to turn the current study into a pilot study. Thank you. The authors renominated the title, however, more than the nomination of the title, the transformations of the text to a pilot study imply a huge change in the content.

As we all know, the purpose of a pilot study in the health sciences is to test and refine research methods and procedures, to assess the feasibility of the research design, and to identify potential challenges, areas for improvement or limitations in the study design, such as logistical issues, data collection methods or selection of study participants, before undertaking a larger study. one can find some gaps between the current text and the purpose of pilot studies: (a) Are the methods being tested or presented as 'conclusive' or do they bring new lessons and suggested changes for application in a significant sample of the spastic cerebral palsy population? (b)  In another complementary way, is the methodology adequate for a study with a larger number of children with the same non-homogeneous characteristics as the eight now presented? (c) Is the equipment approved for a correct measurement of the stance phase of the gait in children with different foot contact behaviours, namely the "foot strike"?

In the context of the above considerations, I dare to withdraw the suggestion of a pilot study, but find a title for the study that is more assertive and that "defends" the authors. A suggestion for a new title could be:

Assessment of five foot plantar morphological pressure points of children with cerebral palsy using or not "dynamic ankle foot orthosis"

==

2. About the authors answer about the " homogeneity of the study"

If "Homogeneity could not be achieved within the group because it was studied with a small group.", on what basis are results from children aged 4 to 12 years mixed? These children are naturally morphologically very different and there is no guarantee that cerebral palsy has affected all children in the same way. So, they walk in different patterns. What statistical evidence has been used and based on what criteria? Table 1 is not explicit.  It prejudices the paper.There is no appendix with all the results. Important for readers. Before drawing conclusions about the group of eight children, it would be good to have access to all the results. For example, in an appendix or in the text, a table showing all the results from the 8 children.

==

3. About "Measurements are made by placing sensors at certain points …" - Agree with the authors.

==

4. About "GMFCS scale" – Thank you for the explanation

With this explanation the reader understands what the authors mean by "mild" cerebral palsy which appears in the text

==

5. About my question "Line 235: What the meaning of "mild centers of gravity" and the answer of the authors: "Chow Tong-Hsien. Traceable Features of Static Plantar Pressure Characteristics and Foot Postures in College Students with Hemiplegic Cerebral Palsy. J. Pers. Med. 2022;12: 394."

Comment. Thank you for bringing this article by Chow to my attention. I read it with great interest. Despite strong criticism of some confusions related to "mild" applied to "centres of gravity" the article is very well structured and a reference to follow if (and only if) it is carefully "corrected" of some conceptual (or is it simply terminological) misunderstandings. At the end please decide whether to keep the "mild" or adjust the conceptual terminology. Sorry my long text. I will talk about concepts and none observation about translation to English:

In my opinion, when we quote other authors, we need to clarify if we agree, disagree or if there is some other reason. I respectfully think that you have not had time to certify all the details of the quoted article. Therefore, I give my opinion after an objective reading of the article by Chow Tong-Hsien. First, but not important for today's "discussion", the paper is about people attending college. Your study is about the performance of children. But more importantly. Something went wrong in Chow´s's paper because there are major conceptual misunderstandings about the scientific specificity of what is "centre of gravity" ...

There is a misunderstanding (which appears in many authors) between centre of gravity (CoG) and one of the two following concepts: "projection on the support (ground) of the centre of gravity (CoGp)" or "centre of support (CoP)". The CoG is located at a height from the support (ground) that depends on the morphology of the performer and his intersegment movements. it has 3D coordinates; the CoP and CoGp have only 2D coordinates measured on the support (ground). The CoP depends on the distribution of the pressure forces at the support (ground). The CoP may be the result of all the supports of the two feet (D. Winter called it CoP_net; or each foot has its own CoP (CoP_right; CoP_left). As is well known, the "CoP_net" has nothing (!) in common with the CoGp. Only by coincidence can they be on the same coordinate. Therefore, CoG, CoGp and CoP are independent and are measured in 3D or a 2D metric coordinates. They can never be "mild". Metric quantities are not "mild" ("gentle", "moderate"). Giving the benefit of the doubt I think Chow every time he uses "centre of gravity" refers to "centre of pressure" (but I've never understood if it's the "CoP_net" or each of the two feet. However as at certain points in the text it designates "mild centres of gravity" I assume it separates the two supports. In 3.3 the paper in focus Chow write "The centers of gravity were presented as a percentage of the gravity." This is another conceptual misunderstanding. The "gravity" is a concept from Physics ... I assume the author would mean "body weight". The pressure values of each of the supports is presented as a percentage of the performer's body weight".

==

6. About my question "Line 186: Precise: “Many studies stated that the increased plantar flexor moment in the stance phase was a biomechanical utility of orthoses in spastic diplegia”.

The authors did not respond. At this point, there is also a terminology or conceptual issue. What is a "biomechanical utility"? Is biomechanical a method of analysis or a noun that can have more or less utility?

Author Response

First of all, I would like to say that each of your comments is very educational for us. Thank you for your comments. The perspective of an experienced person in the field is very important to us. As in every study, we may not be enough at some points, we take care to complete them as much as possible.

The study under review deserves my appreciation. The authors' replies show an effort to respond to the warnings, but the work still needs some fine-tuning, whether in the presentation of the results (a table with all the results is suggested below) or in some conceptual (?) or terminological (?) adjustments that are always difficult for non-native speakers of English to express.

  1. About the new title

The authors accepted the suggestion to turn the current study into a pilot study. Thank you. The authors renominated the title, however, more than the nomination of the title, the transformations of the text to a pilot study imply a huge change in the content.

As we all know, the purpose of a pilot study in the health sciences is to test and refine research methods and procedures, to assess the feasibility of the research design, and to identify potential challenges, areas for improvement or limitations in the study design, such as logistical issues, data collection methods or selection of study participants, before undertaking a larger study. one can find some gaps between the current text and the purpose of pilot studies: (a) Are the methods being tested or presented as 'conclusive' or do they bring new lessons and suggested changes for application in a significant sample of the spastic cerebral palsy population? (b)  In another complementary way, is the methodology adequate for a study with a larger number of children with the same non-homogeneous characteristics as the eight now presented? (c) Is the equipment approved for a correct measurement of the stance phase of the gait in children with different foot contact behaviours, namely the "foot strike"?

In the context of the above considerations, I dare to withdraw the suggestion of a pilot study, but find a title for the study that is more assertive and that "defends" the authors. A suggestion for a new title could be:

Assessment of five foot plantar morphological pressure points of children with cerebral palsy using or not "dynamic ankle foot orthosis"

Thank you for comments.

  1. About the authors answer about the "homogeneity of the study"

If "Homogeneity could not be achieved within the group because it was studied with a small group.", on what basis are results from children aged 4 to 12 years mixed? These children are naturally morphologically very different and there is no guarantee that cerebral palsy has affected all children in the same way. So, they walk in different patterns. What statistical evidence has been used and based on what criteria? Table 1 is not explicit.  It prejudices the paper.There is no appendix with all the results. Important for readers. Before drawing conclusions about the group of eight children, it would be good to have access to all the results. For example, in an appendix or in the text, a table showing all the results from the 8 children.

You are right that each of these children can be different morphologically, we agree with you. We wanted to give a general result. Each child can be interpreted differently as a separate case. We gave the overall averages using the non-parametric statistical test

  1. About "Measurements are made by placing sensors at certain points …" - Agree with the authors.

Thank you for comments.

  1. About "GMFCS scale" – Thank you for the explanation

With this explanation the reader understands what the authors mean by "mild" cerebral palsy which appears in the text

Thank you for comments.

  1. About my question "Line 235: What the meaning of "mild centers of gravity" and the answer of the authors: "Chow Tong-Hsien. Traceable Features of Static Plantar Pressure Characteristics and Foot Postures in College Students with Hemiplegic Cerebral Palsy. J. Pers. Med. 2022;12: 394."

Comment. Thank you for bringing this article by Chow to my attention. I read it with great interest. Despite strong criticism of some confusions related to "mild" applied to "centres of gravity" the article is very well structured and a reference to follow if (and only if) it is carefully "corrected" of some conceptual (or is it simply terminological) misunderstandings. At the end please decide whether to keep the "mild" or adjust the conceptual terminology. Sorry my long text. I will talk about concepts and none observation about translation to English:

In my opinion, when we quote other authors, we need to clarify if we agree, disagree or if there is some other reason. I respectfully think that you have not had time to certify all the details of the quoted article. Therefore, I give my opinion after an objective reading of the article by Chow Tong-Hsien. First, but not important for today's "discussion", the paper is about people attending college. Your study is about the performance of children. But more importantly. Something went wrong in Chow´s's paper because there are major conceptual misunderstandings about the scientific specificity of what is "centre of gravity" ...

There is a misunderstanding (which appears in many authors) between centre of gravity (CoG) and one of the two following concepts: "projection on the support (ground) of the centre of gravity (CoGp)" or "centre of support (CoP)". The CoG is located at a height from the support (ground) that depends on the morphology of the performer and his intersegment movements. it has 3D coordinates; the CoP and CoGp have only 2D coordinates measured on the support (ground). The CoP depends on the distribution of the pressure forces at the support (ground). The CoP may be the result of all the supports of the two feet (D. Winter called it CoP_net; or each foot has its own CoP (CoP_right; CoP_left). As is well known, the "CoP_net" has nothing (!) in common with the CoGp. Only by coincidence can they be on the same coordinate. Therefore, CoG, CoGp and CoP are independent and are measured in 3D or a 2D metric coordinates. They can never be "mild". Metric quantities are not "mild" ("gentle", "moderate"). Giving the benefit of the doubt I think Chow every time he uses "centre of gravity" refers to "centre of pressure" (but I've never understood if it's the "CoP_net" or each of the two feet. However as at certain points in the text it designates "mild centres of gravity" I assume it separates the two supports. In 3.3 the paper in focus Chow write "The centers of gravity were presented as a percentage of the gravity." This is another conceptual misunderstanding. The "gravity" is a concept from Physics ... I assume the author would mean "body weight". The pressure values of each of the supports is presented as a percentage of the performer's body weight".

Thank you for the clarification, we have removed this study from the discussion section

  1. About my question "Line 186: Precise: “Many studies stated that the increased plantar flexor moment in the stance phase was a biomechanical utility of orthoses in spastic diplegia”.

The authors did not respond. At this point, there is also a terminology or conceptual issue. What is a "biomechanical utility"? Is biomechanical a method of analysis or a noun that can have more or less utility?

Many studies have indicated that the increased plantar flexor moment in the stance phase provides some biomechanical benefits during walking to the child with spastic diplegia. They concluded that AFO improves walking function in diplegic CP compared with barefoot walking.’ We changed the sentence. We wanted to point out the biomechanical benefit of the orthosis during gait.

Reviewer 3 Report

Dear authors, 

Thank you for making the changes proposed.

Manuscript looks clearer.

Thank you

Author Response

Thank you for your comments